# Diversity of *Anaplasma phagocytophilum* Strains from Roe Deer (*Capreolus capreolus*) and Red Deer (*Cervus elaphus*) in Poland

**DOI:** 10.3390/ani14040637

**Published:** 2024-02-16

**Authors:** Anna W. Myczka, Żaneta Steiner-Bogdaszewska, Grzegorz Oloś, Anna Bajer, Zdzisław Laskowski

**Affiliations:** 1Department of Eco-Epidemiology of Parasitic Diseases, Institute of Developmental Biology and Biomedical Sciences, Faculty of Biology, University of Warsaw, Ilji Miecznikowa 1, 02-096 Warsaw, Poland; a.bajer2@uw.edu.pl; 2Witold Stefański Institute of Parasitology, Polish Academy of Sciences, Twarda 51/55, 00-818 Warsaw, Poland; zaneta.steiner@gmail.com (Ż.S.-B.); laskowz@gmail.com (Z.L.); 3Institute of Environmental and Engineering and Biotechnology, University of Opole, Kardynała B. Kominka 6, 6a, 45-032 Opole, Poland; golos@uni.opole.pl

**Keywords:** *Anaplasma phagocytophilum*, *groEL*, *ankA*, tick-borne diseases, wildlife

## Abstract

**Simple Summary:**

Bacteria continuously circulate in the natural environment and frequently cross into the human environment. While some of these bacteria can have a positive effect on human health, they are often accompanied by pathogenic forms. The aim of the present study was to more closely examine the zoonotic, tick-borne bacterium *Anaplasma phagocytophilum,* the causative agent of human and animal anaplasmosis. The study examined tissue samples from roe deer and red deer. *Anaplasma phagocytophilum* samples were genotyped based on DNA amplification, sequencing and phylogenetic analysis. Our results showed that detected bacteria from red deer belong to ecotype I and cluster I, which is why those bacteria are potentially pathogenic to humans. Detected bacteria from roe deer were mainly in ecotype II and clusters II and III, which means those bacteria are not pathogenic for humans.

**Abstract:**

Background: The Gram-negative bacterium *Anaplasma phagocytophilum* is an intracellular pathogen and an etiological agent of human and animal anaplasmosis. Its natural reservoir comprises free-ranging ungulates, including roe deer (*Capreolus capreolus*) and red deer (*Cervus elaphus*). These two species of deer also constitute the largest group of game animals in Poland. The aim of the study was to genotype and perform a phylogenetic analysis of *A. phagocytophilum* strains from roe deer and red deer. Methods: Samples were subjected to PCR amplification, sequencing, and phylogenetic analysis of strain-specific genetic markers (*groEL*, *ankA*). Results: Five haplotypes of the *groEL* gene from *A. phagocytophilum* and seven haplotypes of *ankA* were obtained. The phylogenetic analysis classified the *groEL* into ecotypes I and II. Sequences of the *ankA* gene were classified into clusters I, II, and III. Conclusions: Strains of *A. phagocytophilum* from red deer were in the same ecotype and cluster as strains isolated from humans. Strains of *A. phagocytophilum* from roe deer represented ecotypes (I, II) and clusters (II, III) that were different from those isolated from red deer, and these strains did not show similarity to bacteria from humans. However, roe deer can harbor nonspecific strains of *A. phagocytophilum* more characteristic to red deer. It appears that the genetic variants from red deer can be pathogenic to humans, but the significance of the variants from roe deer requires more study.

## 1. Introduction

The Gram-negative bacterium *Anaplasma phagocytophilum* of the order *Rickettsiales* is a common intracellular parasite of both veterinary and medical interest [1,2]. *A. phagocytophilum* has been recorded globally [3,4,5,6,7,8,9,10,11,12,13,14,15,16]. The bacteria are typically transmitted by ticks and mainly by *Ixodes ricinus* in Europe [17,18]. There is currently no evidence of vertical transmission of *A. phagocytophilum* in the tick population; as such, the presence of *A. phagocytophilum* bacteria in the natural environment has most likely resulted from the overlap between the ecological niches of the vectors and animal reservoir hosts [19,20,21]. These bacteria are etiological agents of anaplasmosis in both humans (human granulocytic anaplasmosis, HGA) and animals [22,23]. The symptoms include fever, headache, chills, and gastric problems, with thrombocytopenia, leukopenia, and increased liver enzyme activity being noted in blood tests. However, all these symptoms are nonspecific, and diagnosing anaplasmosis can present a challenge without information about the tick bite [24,25]. HGA is common in North America, mainly in the USA, with over 5500 cases of anaplasmosis diagnosed in 2019; that same year in Europe, only about 300 cases were reported (Centers for Disease Control and Prevention—CDC) [25]. In animals, the occurrence of *Anaplasma phagocytophilum* was first discovered in sheep (*Ovis aries*) in Scotland as *Cytoecetes phagocytophila* [3]. In Europe, *A. phagocytophilum* strains appear to be more virulent among animals than humans [26]. Cattle infected with *A. phagocytophilum* present with high fever, anemia, leukopenia, and thrombocytopenia, as well as lower milk production, weight loss, and dullness. In wild ruminants, infection with *A. phagocytophilum* is asymptomatic and does not significantly affect the condition of the animals. However, in severe disease, the symptoms usually are weight loss, anorexia, and apathy [27,28]. Interestingly, while animal anaplasmosis has been observed in Asia [29,30], Africa [31,32], and Europe [33], no such record exists in North America [34,35]. A few fatal cases due to *A. phagocytophilum* infection have been reported in Norway, among farmed animals (sheep) and wild game animals (roe deer, *Capreolus capreolus*; moose, *Alces alces*) [36,37]. In many European countries, *A.phagocytophilum* was detected in wild and farm cervids [38]. In roe deer and red deer (*Cervus elaphus*), hosts were detected in Norway [37], Poland [39], France [40], Italy [41], Austria, Czech Republic [42], Germany [43], Hungary [44], Spain [45], and Slovakia [46] and additionally genotyped with various genetic markers such as *16S* rDNA, *mps2*, *msp4*, *groEL*, and *ankA*. 

In the research field of animals and humans anaplasmosis, five genetic markers are most common: *16S* rDNA, *groEL*, *ankA*, *msp2,* and *msp4* [1,2,20,26,38,39]. In this study, two genetic markers from this group, *groEL* and *ankA*, were used for the genotyping of *A. phagocytophilum* in two deer species. The *groEL* gene encodes heat-shock proteins, which enable the classification of *A. phagocytophilum* strains into four ecotypes [20,38,39]. The ecotypes of *A. phagocytophilum* clustered strains can occur in the similar or identical ecological niche in various hosts [39]. Variants I and II can infect a wide range of hosts: ecotype I is associated with goats (*Capra hircus*), hedgehogs (*Erinaceus europaeus*), roe deer, and humans, while ecotype II is mainly associated with wild ruminants—roe deer and moose. Ecotypes III and IV are more host-specific; i.e., ecotype III infects rodents, and ecotype IV infects birds [38,40]. All four ecotypes have been found in tick vectors [38,47,48]. The *ankA* gene encodes a cytoplasmic antigenic protein, whose sequence allows *A. phagocytophilum* strains to be divided into five clusters, each one associated with a certain host species [26,49,50]. Cluster I includes strains isolated from various hosts: humans and animals (domestic, wild, and farmed). Cluster II and III are formed by unique *A. phagocytophilum* genetic variants from roe deer and are probably not pathogenic to humans or other animals. Cluster IV is associated mainly with ruminants, e.g., sheep, European bison (*Bison bonasus*), and cows (*Bos taurus taurus*). Cluster V represents *A. phagocytophilum* variants circulating among species, namely rodents [26,40,49,50,51].

The aim of this study was to genotype *A. phagocytophilum* strains from two game species, viz. red and roe deer, with the use of *groEL* and *ankA* partial genetic markers. It focuses on the classification of *A. phagocytophilum* into ecotypes and clusters. The results of the phylogenetic analysis of the detected strains from red and roe deer can be used to determine which ones may, or may not, be potentially pathogenic to humans.

## 2. Materials and Methods

### 2.1. Materials

In total, 160 spleen and liver samples were collected from free-living and farmed deer. In 2017–2020, 145 samples were collected during hunting season in four areas: Pisz Forest (Warmian-Masurian Voivodeship), Bolimów Forest (Łódź Voivodeship), Kampinos National Park (Masovian Voivodeship), and Stobrawa-Turawa Forest (Opolskie Voivodeship). In addition, 15 spleen and liver samples were organized from farmed red deer from the Research Station of Witold Stefański Institute of Parasitology, Polish Academy of Sciences in Kosewo Górne (Warmian-Masurian Voivodeship). The samples are organized by host species, sex, and age in Table 1. All collected samples were previously examined for *Anaplasma* spp. With the use of the nested PCR amplification of *16S* rDNA gene [52]. From the studied group of cervids, samples from 50 individuals of red deer and 39 individuals of roe deer were genotyped with the use of *groEL* and *ankA* genetic markers.

### 2.2. Methods

DNA was isolated from spleen and liver samples using a commercial DNA Mini Kit (Syngen, Poland) according to the manufacturer’s protocol. Positive samples for *Anaplasma* spp. were genotyped via the nested PCR amplification of two markers, *groEL* and *ankA*, according to Alberti et al. [53] and Massung et al. [54], respectively. Since the amplification of the *ankA* partial gene from roe deer *A. phagocytophilum*-positive samples was unsuccessful, new primers were developed at the Department of Ecology and Evolution of Parasitism of the Witold Stefański Institute of Parasitology, Polish Academy of Sciences (Table 2). The nested PCR amplification of the *ankA* gene with the new primers was performed according to Massung et al. [54]. For the new primers, a mix of two forward and one reverse primer was used in the first reaction. In the second reaction, a mix of two reverse and one forward were used for the amplification of a ~670-bp of the *ankA* gene. Amplification of the *groEL* partial gene was performed using the DNA Engine T100 Thermal Cycler (BioRad, Hercules, CA, USA).The first reaction was performed according to the following program: denaturation at 95 °C for two minutes, followed by 34 cycles of denaturation at 95 °C for 10 s, annealing at 50 °C for 15 s and extension at 72 °C for 30 s, with a final extension performed at 72 °C for five minutes. The second reaction was performed according to the following program: denaturation at 95 °C for one minute; followed by 34 cycles of denaturation at 95 °C for 10 s, annealing at 55 °C for 15 s, and extension at 72 °C for 30 s, with a final extension performed at 72 °C for three minutes. DNA amplification of the *ankA* partial gene was performed using the DNA Engine T100 Thermal Cycler (BioRad, Hercules, CA, USA). The first reaction was performed according to the following program: denaturation at 95 °C for three minutes, followed by 34 cycles of denaturation at 95 °C for 10 s, annealing at 50 °C for 10 s, and extension at 72 °C for 45 s, with a final extension performed at 72 °C for five minutes. The second reaction was performed according to the following program: denaturation at 95 °C for one minute, followed by 34 cycles of denaturation at 95 °C for 10 s, annealing at 55 °C for 20 s and extension at 72 °C for 30 s, with a final extension performed at 72 °C for three minutes. All the reactions were conducted in a 20 μL reaction mixture volume containing 2 μL of DNA, 0.1 μL (5U) of Gold Taq Polymerase (Syngen, Wrocław, Poland), 0,2 μL of dNTPs mix (25 mM) (Syngen, Poland), 1 μL of each primer (10 mM), 2 μL of polymerase buffer (Syngen, Poland), and 13,7 μL of nuclease-free water. The DNA of *A. phagocytophilum* isolated from *Alces alces* was used as a positive control (*groEL*—OM835684, *ankA*—OM835678) [16]. A negative control, consisting of nuclease-free water, was also added to the PCR mix instead of the tested DNA. The PCR products were visualized on a 1.2% agarose gel (Promega, Madison, WI, USA) stained with SimplySafe (EURx, Gdańsk, Poland) and a size-marked DNA Marker 100 bp LOAD DNA ladder (Syngen, Poland). Visualization was performed using ChemiDoc, MP Lab software (Imagine, BioRad, USA). The obtained PCR products were purified with the DNA clean-up Kit (Syngen, Poland), sequenced with the Sanger method by Genomed (Warsaw, Poland), and assembled using ContigExpress, Vector NTI Advance v.11.0 (Invitrogen Life Technologies, Carlsbad, CA, USA). The obtained sequences were compared with the GenBank database using BLAST (NCBI, Bethesda, MD USA) and then submitted to GenBank. Phylogenetic trees were constructed using Bayesian inference (BI), as implemented in the MrBayes version 3.2.0 software [55]. The HKY + I model was selected for the *groEL* sequences and the GTR + I + G model for *ankA* sequences as the best-fitting nucleotide substitution models, using JModelTest version 2.1.10 software [56,57]. The analysis was run for 2,000,000 generations, with 500,000 generations discarded as burn-in. The phylogenetic trees were visualized using the TreeView software (S&N Genealogy Supplies, Chilmark, Salisbury, Wiltshire, UK).

## 3. Results

### 3.1. GroEL Diversity

Five unique genetic variants (haplotypes) were identified among the twelve obtained sequences of the *groEL* partial gene fragment, including six from the red deer and six from the roe deer. The obtained sequences demonstrated 100% identity to the *groEL* sequences obtained from red deer in many European countries (listed in Table 3). The six sequences of the *A. phagocytophilum groEL* gene fragment obtained from red deer were identical (ON604852–ON604857) and classified as ecotype I. However, the six *groEL* sequences obtained from roe deer were classified into four genetic variants (haplotypes) (ON604846, ON604848, ON604849, and ON604850). The first haplotype group (G1) was represented by three sequences, namely ON604846, ON604847, and ON604851, while each remaining group included a single haplotype: ON604848 in GII, ON604849 in GIII, and ON604850 in GIV. These sequences demonstrated 100% identity to the *groEL* sequences obtained from roe deer in many European countries (Table 3). Haplotypes from group I and II belonged to ecotype II; haplotypes from group III and IV represented ecotype I (Figure 1).

### 3.2. AnkA Diversity

Seven haplotypes were identified among the ten obtained *ankA* sequences. Of these, four haplotypes (ON646026, ON646029, ON646030, and ON646031) were identified among six *A. phagocytophilum* sequences from roe deer. One haplotype encompassed three identical sequences (ON646026, ON646027, and ON646028). Three haplotypes of the *A. phagocytophilum ankA* gene fragment reported in this study (ON646026, ON646029, and ON646030) had already been reported in the GenBank database. The fourth haplotype, i.e., ON646031, had no 100% identity with any submission in GenBank (Table 4). Three haplotypes were identified among the four *ankA* sequences obtained from red deer. One of these haplotypes was represented by two identical sequences (ON646033 and ON646034) from two different individual red deer, while the other two demonstrated only one sequence each (ON646032 and ON676564). None of the red deer *ankA* gene variants displayed 100% similarity with GenBank records (Table 4). The phylogenetic analysis indicated that the sequences obtained from roe deer belonged to cluster I, II, and III, and all sequences from the red deer belonged to cluster I (Figure 2).

## 4. Discussion

The circulation of *Anaplasma phagocytophilum* in the natural environment among wild game animals is studied and monitored as part of the WHO’s “One Health” program. The balance referred to in the “One Health” program is threatened by the presence of a large and uncontrollable reservoir of *Anaplasma* genus bacteria, which in Poland consists of cervids. Therefore, our present study focused on the two cervid species, red deer and roe deer, which happen to be two of the largest populations of wild game animals in the country. Our findings based on the use of two genetic markers, *groEL* and *ankA*, provide a deeper insight into the diversity of *A. phagocytophilum* strains and allow us to better understand the nature of the parasite and its potential threat to human and animal health. 

Our findings confirm the presence of a range of genetic haplotypes of *Anaplasma phagocytophilum* in both species of deer, including possible zoonotic strains from ecotype I and cluster I. They also confirm an association between certain haplotypes of *A. phagocytophilum* with roe deer (ecotypes II and clusters II and III) and red deer (ecotype I and cluster I). The roe deer were found to harbor one new haplotype of the *ankA* gene involved in cluster I; this phylogenetic finding is the first record of *A. phagocytophilum* from roe deer associated in cluster I [26].

Five haplotypes of the *groEL* sequence were identified among the deer. An analysis of the *groEL* sequences in BLAST showed that all these haplotypes can be classified as European ecotypes. The obtained haplotypes did not demonstrate any high similarity (90–95%), with any of the *groEL* sequences present in the GenBank database from Asia, MT010579.1, MN989864 [58], MH722253, MH722254 [59], KY379956; North America, AF383227 [60], AY219849 [61], JF494840 [62], DQ680012; and Middle America, MW699686, MW699687 [63]. Interestingly, in red deer, only one haplotype of *groEL* was recorded, which was found to belong to ecotype I. This genetic variant has quite a broad specificity and has previously been identified in a wide range of hosts, including red deer [48], horses (*Equus caballus)* [64], sheep [65], and ticks [48,66,67]. In samples from roe deer, four groups of *groEL* haplotypes were obtained (group I: ON604846, ON604847, ON604851, group II: ON604848, group III: ON604849, group IV: ON604850). The haplotypes from groups I and II belong to ecotype II, and those from groups III and IV belong to ecotype I (Figure 1). Ecotype I, as mentioned above, is associated with humans and domestic, farm, and wild game animals [38,48,64,65,66,67]. While sequences of *A. phagocytophilum* ecotype II are commonly reported throughout Europe [38,68,69], their occurrence seems restricted only to three host species: roe deer, moose, and the *Ixodes* genus ticks (Table 3). In Poland, among wild animals, previously analyzed strains of *A. phagocytophilum* have been found to occupy ecotypes I or II [16,39,68,69,70], which is in line with our findings. In addition, the fact that the obtained partial *groEL* gene haplotypes from red and roe deer are present in ecotype I and those from roe deer are in ecotype II is in agreement with previous reports on *A phagocytophilum* ecotype distribution [37].

Our findings confirm the presence of seven genetic variants of the *Anaplasma phagocytophilum ankA* sequence, belonging to clusters I, II, and III. Cluster I, which includes strains isolated from various hosts, encompassed samples from all red deer and one roe deer. The more host-specific clusters II and III included only the *A. phagocytophilum ankA* haplotypes from roe deer (Figure 2). All these findings are consistent with previous reports [26]. Three of the haplotypes isolated from roe deer (ON646026, ON646029, and ON646030) were recorded in three European countries, viz. Spain, Germany, and Slovenia, but only in roe deer (Table 4) [26] and in ticks of *Ixodes* genus (AY282386) [50]. Previous phylogenetic analyses of the *A. phagocytophilum ankA* sequences identified a strong association between clusters II and III and roe deer [26,40,49,50,51]. One haplotype of the *ankA* gene identified in the present study (ON646031—roe deer isolate S36) is new and did not share 100% identity with any sequence from the GenBank database. The closest match of this haplotype (99.83%) was observed with the *ankA* haplotype from red deer in Slovenia (GU236718) (Table 4) [26]. This haplotype was placed in cluster I (Figure 2), which also encompassed sequences obtained from red deer, humans, and domestic animals such as cats (*Felis catus)* and dogs (*Canis lupus familiaris)* [26]. A further analysis of the *ankA* and *groEL* gene fragments for the *A. phagocytophilum* isolate from roe deer S36 suggests that this individual was a carrier of an *A. phagocytophilum* strain not indicating an association with roe deer. Phylogenetic analysis of the *ankA* sequences from red deer classified them as cluster I; this result is in line with previous findings indicating it to be associated with various hosts, including humans, ticks, game animals like red deer and wild boar (*Sus scrofa*), domestic animals (cats, dogs), and horses [26,40,49,50,51,69]. The haplotype of *A. phagocytophilum* from red deer (ON646033) demonstrated the highest similarity (99.69%) with an *ankA* gene sequence from dogs [64], *I. ricinus* [71], horses [26], sheep [26], and humans [26,49]. The other two haplotypes, ON646032 and ON676564, were not identical but show close similarity with the same sequence. Haplotype ON646032 was 100% identical to GenBank sequence GU236718 obtained from red deer in Slovenia [26]; the second haplotype (ON676564) was found to share 99.68% similarity with the same sequence (Table 4). In Poland, studies that use *ankA* genetic marker are less frequent than those using *groEL* as a marker; however, in the few reports that are available, and in the present study, the *A. phagocytophilum* stains identified in wild animals were assigned to previously identified clusters: red deer and wild boars to cluster I, roe deer to clusters II and III, and bovids to cluster IV [16,26,68,69,72].

Our findings extend existing knowledge on the genetic diversity and host specificity of *Anaplasma phagocytophilum* strains in red and roe deer, the two most common species of wild game animals in Poland. Our phylogenetic and sequence analysis of the *groEL* and *ankA* gene fragments, as well as previous studies based on *16S* rDNA [46], indicate that *A. phagocytophilum* sequences obtained from red deer are identical to *A. phagocytophilum* HGA variants identified in humans or are grouped within the same ecotypes and clusters. On the other hand, most of the haplotypes obtained from roe deer in this study seem host-specific and do not show any similarities to zoonotic strains according to our analysis based on *groEL* and *ankA* or on *16S* rDNA [46]. However, our results indicate that roe deer can also be hosts for less typical *A. phagocytophilum* variants; this unusual haplotype was classified as cluster I, like haplotypes from red deer samples and humans. All these results show that a highly diverse range of *A. phagocytophilum* strains are present in cervids in Poland and are potentially zoonotic. 

## 5. Conclusions

In Poland, roe deer are not likely to be natural reservoirs of HGA strains, but red deer can serve as reservoirs hosts of zoonotic strains. Also, we describe the possible occurrence of a nonspecific strain of *A. phagocytophilum* in roe deer: an *A. phagocytophilum* variant more characteristic of red deer. According to all of these findings, both wild hosts—*Cervus elaphus* and *Capreolus capreolus*—can likely serve as reservoirs of zoonotic *Anaplasma phagocytophilum* strains.

## Figures and Tables

**Figure 1 animals-14-00637-f001:**
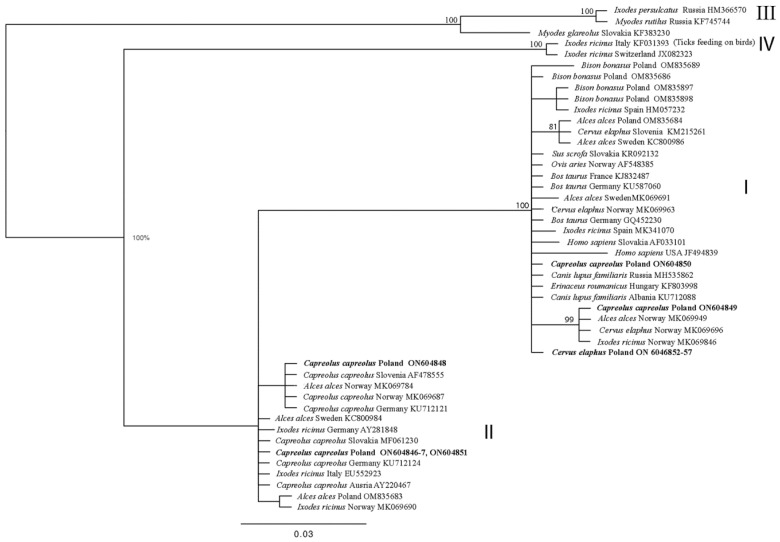
Phylogenetic tree of *groEL* partial gene (486 bp) of *Anaplasma phagocytophilum* haplotypes, constructed using Bayesian inference (BI) analysis with MrBayes version 3.2 [55]. The HKY + I model was chosen as the best-fitting nucleotide substitution model using JModelTest version 2.1.10 software [56,57]. The analysis was run for 2,000,000 generations, with 500,000 generations discarded as burn-in. In bold are sequences from this study. I–IV: ecotypes of *A. phagocytophilum* according to Jahfari et al. [38].

**Figure 2 animals-14-00637-f002:**
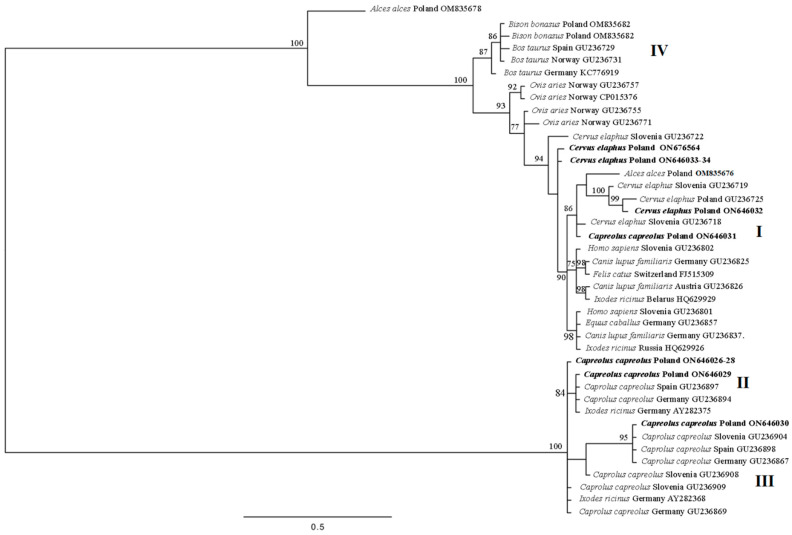
Phylogenetic tree of the *ankA* partial gene (633 bp) haplotypes from *Anaplasma phagocytophilum*, constructed using Bayesian inference (BI) analysis with MrBayes version 3.2 [55]. The GTR + I + G model was chosen as the best-fitting nucleotide substitution model using JModelTest version 2.1.10 software [56,57]. The analysis was run for 2,000,000 generations, with 500,000 generations discarded as burn-in. Sequences from this study are in bold. I–IV: clusters of *A. phagocytophilum* according to Scharf et al. [26].

**Table 1 animals-14-00637-t001:** Demographic characteristics of the study population. The numbers of farmed cervids presented in parentheses.

Species	Adults		Juvenile	Total
	Males	Females		
Red deer	8 (0)	49 (15)	33 (0)	90 (15)
Roe deer	7	49	14	70
In total	15	98	47	160

**Table 2 animals-14-00637-t002:** Primer designed and used to amplify *ankA* gene fragment from roe deer samples.

Reaction	Primers	Tm (°C)	Reference
PCR	ankAF1a 5′-TGCTGTAAATGAAGAAATTACAACTTC-3′ ankAF1b 5′-TGGTGTAAATGAAGAAATTACAACTC-3′ ankARC 5′-GCCTTTAGTAGTACTCTACATGC-3′	53 °C 52 °C 54 °C	this study
Nested—PCR	ankAF2a 5′-CTGACCGCTGAAGCACTAA-3′ ankAR1a 5′-GAAGCCAGATGCAGTAACGA-3′ ankAR1b 5′-GAAGCAAGATGCAGTAACGA-3′	51 °C	
		52 °C	
		50 °C	

**Table 3 animals-14-00637-t003:** Haplotypes of the *A. phagocytophilum groEL* gene fragment and reference sequences from the GenBank database, divided into ecotypes. Sequences from this study in bold.

Ecotype		Host		No. GenBank Sequence (This Study)	Sequences with 100% Similarity	Host	Country
I	red deer (*Cervus elaphus*)		**ON604852-87**		MK069963	red deer	Norway
					KU712106		Austria
					KJ832471	horse	France
					MZ348280	sheep	Germany
					MK069889	*Ixodes ricinus*	Norway
					MW732493		UK
					KF312358		Poland
	roe deer (*Capreolus capreolus*)		**ON604849**		MK069949	moose	Norway
					MK069696	red deer	
					MK069797	*I. ricinus*	
			**ON604850**		AY281823	*I. ricinus*	Germany
					KJ832474	cow	France
					GQ452227	goat	Switzerland
					MW013536	hedgehog	Czech Republic
II	roe deer (*Capreolus capreolus*)		**ON604846-47** **ON6048451**		MN093177	*I. ricinus*	Nederland
					KU712112	roe deer	Germany
					KC800984	moose	
					AY220467	roe deer	Austria
			**ON604848**		KF031380	*I. ricinus*	Italy
					HQ629905		Estonia
					MK069774	moose	Norway
					GQ988754	roe deer	Austria
					KU712121		Germany

**Table 4 animals-14-00637-t004:** Haplotypes of *A. phagocytophilum ankA* gene fragment and reference sequences from the GenBank database, divided into clusters. Sequences from this study in bold.

Cluster	Host	No. GenBank Sequence (This Study)	Sequences with Highest Identity	Host	Country
I	red deer (*Cervus elaphus*)	**ON646032**	GU236718 (100%)	red deer	Slovenia
		**ON646033-34**	KJ832286 (99.69%)	dog	France
			HQ629928 (99.69%)	*Ixodes ricinus*	Estonia
		**ON676564**	GU236718 (99.68%)	red deer	Slovenia
	roe deer (*Capreolus capreolus*)	**ON646031**	GU236718 (99.83%)	red deer	Slovenia
			KJ832286 (99.66%)	dog	France
			HQ629928 (99.66%)	*Ixodes ricinus*	Estonia
II	roe deer (*Capreolus capreolus*)	**ON646026-28**	GU236909 (100%)	roe deer	Slovenia
			GU236894 (100%)		Germany
			GU236900 (100%)		Spain
			AY282386 (100%)	*Ixodes ricinus*	Germany
		**ON646029**	GU236897 (100%)	roe deer	Spain
			GU236874 (100%)		Germany
			AY282375 (100%)	*Ixodes ricinus*	
III	roe deer (*Capreolus capreolus*)	**ON646030**	GU236904 (100%)	roe deer	Slovenia
			GU236898 (100%)		Spain

## Data Availability

The data that support the findings of this study are available from the corresponding author upon reasonable request.

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
