# Peer review of "Diversity of Anaplasma phagocytophilum Strains from Roe Deer (Capreolus capreolus) and Red Deer (Cervus elaphus) in Poland"

_animals, 2024, doi:10.3390/ani14040637_

Round 1

Reviewer 1 Report

Comments and Suggestions for Authors

Dear Authors,

Thank you for submitting this valuable piece of knowledge about the natural occurence of Anaplasma phagocytophilum haplotypes in competent hosts. The manuscript needs some language editing before being acceptable for publication.

Look particularly at word orders, pronouns, definite articles, plurals and singulars. 

Please also check table 1. The row denoted "roe deer" has a number attached to it.

Otherwise, I see no suggestions for amendments.   

Comments on the Quality of English Language

The English quality must be improved. The word order and grammar have several issues throughout the text. 

Author Response

Dear Reviewer,
Authors would like to thank the Reviewer for read the manuscript. We are very glad and grateful that our research reach point where prepared manuscript do not need many improvements.
We send to you an improved manuscript, with suggestion from all Reviewers. Thank to you and your commitment to prepare review this article has significantly increased in value and presents precise and reliable information about the research that we conducted.

Kind regards
Anna W. Myczka (Correspondent Author)

Reviewer 2 Report

Comments and Suggestions for Authors

L18-L19 it is too complicate for the simple summary, it is not clear what is a cluster and ecotype, and it is important that certain ecotype and cluster was found. Please rephrase that it would be clear for broader readership. Are these clusters and ecotypes important for humans

L24 roe deer and red deer

L28 delete “(genetic variants)”, it is obvious

L29 it should be “haplotypes, within”

L30 by phylogenetic analysis

According to requirements “Abstract should be a total of about 200 words maximum”. Now it is 212. L34-35 the meaning of this sentence is confusing.

L47 An explanation of the HGA abbreviation is required

L40-60 make a separate paragraph on anaplasmosis in animals and expand it a bit

Introduction: you should improve it and modify/expand in response to following questions. 1) What other genetic markers besides groEL and ankA were used or not used to genotype A. phagocytophilum for the understanding epidemiology which ecotypes, clusters are related to certain hosts 2) what are the difference between ecotypes and clusters 3) how many similar investigations examining A. phagocytophilum in members of the family Cervidae were performed in Europe, and exactly in which cervid species – it allow to understand whether the current study is important for Poland or for whole Europe.

L91-93 briefly, by what methods A. phagocytophilum were examined, identified.

L109-110 Table 2

L134-135 I do not understand methodology, nested PCR requires four priemrs, semi-nested PCR three primers. Three primers each are given for PCR and nested PCR, so what is the difference? Also, authors mentioned that they used nested PCR, so why PCR also is indicated in Table 2? What is length of amplified fragments of both genes? Please include all details of PCR in the text, now it is unclear.

L118 more details on sequencing are needed

L113 only water, or PCR mix with water instead of DNA?

L141, L149, L182, L231, L243, L249, L262 Table

L145-L150 Please rewrite, term “haplotype groups” is rebundant and confusing. “haplotype groups” are not indicated in Figure 1.

L171 delete “genetic”

Not Fig.1. ans 2 but Figure 1. and 2; please correct

L171 Why only 10 sequences were obtained comparing with 12 sequences of the another gene?

L174 it has not be shown in the Table 4, i.e. identity of ON646026-ON646028, the same is for ON646033 and ON646034

L204-205 please include reference

The only novelty of the study that in single strain of roe deer I ecotype of GroEL and I cluster of AnkA was defined. These ecotypes/clusters are pathogenic for humans and for the first time they were identified in roe deer. Are these results from the same strain? What other results obtained in current study are important for the research field on Anaplasma?

Author Response

Dear Reviewer,
Thank you very much for the fact that you all have done so much work and devoted a lot of attention while reviewing our manuscript.
Authors would like to thank the Reviewer for all remarks, comments, recommendations which we could consider with the entire authoring team subscribing to this work. We tried, to the best of our ability, to apply and consider all remarks and comments that you have sent to us. We hope that our answers will satisfy you.
We send to you an improved manuscript, which has significantly increased in value and presents precise and reliable information about the research that we conducted.
Kind regards
Anna W. Myczka (Correspondent Author)

Reviewer 3 Report

Comments and Suggestions for Authors

This manuscript entitled ‘ Diversity of Anaplasma phagocytophilum strains from roe deer 2 (Capreolus capreolus) and red deer (Cervus elaphus) in Poland’’ presents a survey on the diversity of A. phagocytophilum strains in free-ranging ungulates from Poland. In the current study, a total of 160 deer spleen and liver samples from free-living and farm deers were screened for the presence of Anaplasma spp., and the positive samples were genotyped by nested-PCR amplification and sequencing of two markers (groEL and ankA). The findings showed that five genetic variants within groEL gene sequences and seven haplotypes within ankA sequences were detected. The sequences of groEL classified into ecotypes I and II in phylogenetic analysis, while the sequences of ankA gene represented clusters I, II and III. The manuscript is well-organized, and well-written, however, the minor limitation of the manuscript is that:

-The material method section should be more descriptive, for example, how many samples were haplotype analysis performed on?

-It is stated that five unique genetic variants (haplotypes) were identified among 12 obtained sequences of the groEL partial gene fragment. Have all groEL positive samples been sequenced?

Author Response

(The authors gave the same response as above.)

Round 2

Reviewer 1 Report

Comments and Suggestions for Authors

Dear Authors,

Thank you for improving the manuscript. There are still some language issues and some sentences that need revision. The research part is good.

Line 16: roe- and red deer.

Line 18: Should read: Our results showed that detected bacteria from red deer belong to ecotype and cluster I, which means that those bacteria are potentially pathogenic to humans.

Line 19: should read: Detected bacteria from roe deer were mainly of ecotype II and clusters II and III, which means that those bacteria are not pathogenic to humans. 

Line 23: human- and animal granulocytic anaplasmosis

Line 26: roe- and red deer.

line 28: of the groEL gene from Ap

Line 32: different from those isolated from red deer and these strains do not show similarity to bacteria isolated from humans.

Line 58: present with

Line 63: drop there

Line 65: farmed- and wild game animals

Line 66: In the research field ...., five genetic markers

Line 86: species, namely

Line 100: were organised

Line 108: genotyped with the use of

Line 121-128: It is difficult to understand the meaning of the sentences.

Line 112-140: Need language revisions

Line 145 Sanger

Line 170: space

Line 174: Viz?

Line 289: The sentence should be divided or rewritten

Line 295: previously analyzed strains of...

Line 307: the Ixodes genus

Line 319: Indicatin an association with

Line 327: with the use of 

Line 328: than those using groEL

Line 350 reservoirs

Comments on the Quality of English Language

There are some errors and phrasing that need revisions. Please see my suggestions. There are also minor issues, other than those commented on, so please go through the manuscript carefully to improve the language and phrasing.

Author Response

(The authors gave the same response as above.)

Reviewer 2 Report

Comments and Suggestions for Authors

Authors have improved their manuscript, but a number of questions still remain regarding this study

1.     L18-19 Should be “ecotype I and cluster I”

2.     L20-21 rephrase “which mean”, the same wording is used in the sentence above

3.     L55, L60, L116, L170, L261, L268, L333, L350 A. phagocytophilum

4.     L66 the numbers from one to nine must be in letters “five”

5.     L120-137 how many microliters of DNA solution was taken in the first nested PCR step and how many microliters of sample were taken in the second nested PCR step?

6.     Please describe in methods which polymerase was used, what was the concentration of primers in the first and second nested PCR runs or did they differ, and how much of the other PCR components were sampled, in what volume was the PCR performed.

7.     The main novelty of the study that in single strain of roe deer I ecotype of GroEL and I cluster of AnkA was defined. What other results obtained in current study are important for the research field on Anaplasma phagocytophilum? Please include this information in manuscript.

8.     My previous comment was that authors should in INTRODUCTION clarify “how many similar investigations examining A. phagocytophilum in members of the family Cervidae were performed in Europe, and exactly in which cervid species” authors stated that this information was included in the manuscript, I am sorry, but I could not find where exactly, please specify.

Comments on the Quality of English Language

English was improved. Please corect: L20-21 rephrase “which mean”, the same wording is used in the sentence above

Author Response

(The authors gave the same response as above.)

Round 3

Reviewer 2 Report

Comments and Suggestions for Authors

L26-27 Please include Latin names of roe deer and red deer

L67 Please include Latin names of host species (sheep, roe deer and moose) for the first mentioning in the text

L69 Please include Latin name of red deer

L81 delete Capreolus capreolus

L90 delete Ovis aries

L57, L62, L68, L127, L185, L217, L268, L277, L283, L286, L308, L317, L351, L368 A. phagocytophilum should be not Anaplasma phagocytophilum

Author Response

Dear Reviewer,

We send to you an improved manuscript, with suggestion from all Reviewers. Thank to you and your commitment to prepare review this article has significantly increased in value and presents precise and reliable information about the research that we conducted.

Kind regards 

Anna W. Myczka (Correspondent Author)
